# Education for Sustainable Development and Climate Change Education: The Potential of Social Network Analysis Based on Twitter Data

**Alexandra Goritz** [1], **Nina Kolleck** [2,*] and **Helge Jörgens** [3]

1    Department of Educational Research and Social Systems, Freie Universität Berlin, Habelschwerdter Allee 45, 14195 Berlin, Germany; alexandra.goritz@fu-berlin.de
2    Faculty of Social Sciences and Philosophy, Universität Leipzig, Beethovenstraße 15, 04107 Leipzig, Germany
3    ISCTE-University Institute of Lisbon and CIES-Centre for Research and Studies in Sociology, Avenida das Forças Armadas, 1649-026 Lisboa, Portugal; helge.jorgens@iscte-iul.pt
*    Correspondence: nina.kolleck@uni-leipzig.de

**Abstract:** Education is considered an essential tool for achieving sustainability-related goals. In this regard, education for sustainable development (ESD) and climate change education (CCE) have become prominent concepts. The central characteristics of both concepts influence the non-hierarchical network governance structure that has formed around them: (1) their international origin, (2) the conceptual ambiguity that surrounds them, and (3) the limited implementing power of international organizations who developed these concepts. Hence, networks are essential to ESD and CCE, however, only few studies have used social network analysis (SNA) techniques to analyze their governance structure. The aim of this article is to illustrate how to use SNA, based on Twitter data, as an approach to examine the governance structure that has developed around ESD and CCE. We conduct an illustrative SNA, using Twitter data during three global climate change summits (2015-2017) to examine CCE-specific debates and identify actors exerting the most influence. We find that international organizations and international treaty secretariats are most influential across all years of the analysis and, moreover, are represented most often. These findings show that using SNA based on Twitter data offers promising possibilities to better understand the governance structure and processes around both concepts.

**Keywords:** education for sustainable development (ESD); climate change education (CCE); social network analysis (SNA); Twitter; network governance; international organizations; United Nations Educational, Scientific, and Cultural Organization (UNESCO); United Nations Framework Convention on Climate Change (UNFCCC)

## 1. Introduction

Since the beginning of the debate on sustainable development with the so-called "Brundtland Report" in 1987 [1], education has been considered a crucial tool for achieving sustainability-related goals. Different sustainability-related concepts of education were developed over the past years. Among the most popular examples are the concepts of education for sustainable development (ESD) and climate change education (CCE), which have been implemented by several international organizations (IOs) and by national governments around the world. However, debates surrounding the implementation of both concepts are heavily confronted with conceptual disputes and ideological quarrels [2]. One reason for these debates could be the fact that both ESD and CCE evolved from conceptions of environment-related education, which had a narrow focus on environmental issues, to more encompassing concepts aimed at empowering people to become active citizens and enabling them to approach issues of sustainability [3]. ESD and CCE are not only oriented towards the

environmental dimension, but also consider the economic and social dimensions of sustainable development. ESD, on the one hand, can be understood as the "integration of sustainability into all aspects of teaching and learning, in both formal and non-formal education as well as in school curricula" [4] (p. 308). CCE, on the other hand, can be defined as an education that helps develop an adequate response to climate change, increase public awareness and resilience, and empowers people to change their attitudes and behaviors in order to adopt a more sustainable lifestyle [5]. Whereas ESD is a broad concept that covers a wide range of policy areas, CCE focuses more specifically on the challenge of climate change. CCE, therefore, can be conceptualized as a subfield of ESD.

The implementation of ESD and CCE at the global, national, and regional levels requires a set of actors who are interlinked through social relations in a respective governance structure. Three aspects influence this governance structure: First, both concepts have been developed at the international level, mainly by international organizations or within international treaty systems, such as the United Nations Educational, Scientific, and Cultural Organization (UNESCO) and the United Nations Framework Convention on Climate Change (UNFCCC). Thus, a discrepancy persists between the actors and organizations who developed these concepts and those actors at the national and subnational levels who are primarily responsible for their implementation [2]. Second, the concepts of ESD and CCE are still being debated, turning their content rather vague. On the one hand, this leaves implementing actors with a lot of room to use these concepts in accordance with local circumstances. On the other hand, this dilutes the concepts even further. Third, and as a consequence of the former two aspects, the implementation of ESD and, more recently, CCE is progressing only slowly, leaving their potential to promote actual change far from being exhausted. These aspects influence the governance structure around ESD and CCE, which can best be described as a non-hierarchical network in which state and non-state actors together define, promote, and implement ESD as well as CCE [2].

Despite the central role that social networks play for the success of concepts such as ESD, techniques of social network analysis (SNA) have rarely been used to better understand the governance structures and processes that surround them. In this article, we argue that SNA provides scholars with an adequate set of tools, methods, and concepts to study network-like governance structures [4]. Analytically, SNA focuses on the interactions between actors and the relational structures within a network that result from these interactions. This allows to obtain information that would not have been observable otherwise. For example, SNA enables researchers to identify the most influential actors within a governance network in an indirect manner, that is, without the need to directly observe this influence. This represents a departure from previous studies, which have been predominantly based on direct observations or on the (self-)perceptions of actors. SNA thus reduces the risk of systematically excluding potentially influential actors, or of overstating the role and influence of specific actors such as, for example, educational organizations and governmental organizations [6,7]. Hence, in this article, we will illustrate how SNA can be used to study the governance structure that has evolved around the concept of ESD and identify the most influential actors in this process. Our exemplary SNA will be based on data from the online social network (OSN) Twitter. This data source has not been used a lot in the context of ESD, although it can offer important insights into the structure and workings of governance networks. The platform provides political (state and non-state) actors with the opportunity to communicate directly with each other, but also with individual citizens. Interactions between these actors located on different levels become directly observable.

In order to show how SNA, based on Twitter data, can be used to examine the governance structure and ESD-related information exchange, the article is divided into five sections. Section 2 presents the concept of ESD and explains its specificity. Subsequently, we make the case for using a network approach for studying ESD and introduce different possibilities to conduct SNA. Section 4 focuses on Twitter data as a source for SNA, its advantages, and the possibilities to obtain this kind of data. We then conduct an illustrative SNA on the topic of CCE as a subfield of ESD with Twitter data obtained during the global climate negations over three years (2015–2017) and debate the limitations of Twitter data. Finally, we discuss our results and show future research directions.

## 2. The Concept of Education for Sustainable Development

The concept of ESD dates back to the publication of the World Commission on Environment and Development's report "Our Common Future", better known as the Brundtland Report, in 1987 [1]. It was strengthened in Agenda 21, the United Nations Programme of Action adopted at the 1992 United Nations Conference on Environment and Development (UNCED) in Rio de Janeiro [8]. The term ESD links the idea of sustainable development to the much older concept of environmental education [9]. With the proclamation of the decade of 2005–2014 as the 'United Nations Decade of Education for Sustainable Development' [10], (see also [2,11]), the concept of ESD has evolved to become one of the most influential normative ideas at the intersection of environmental and education policies. Beyond the decade of ESD, the concept is currently promoted through the Global Action Programme (GAP) on Education for Sustainable Development [12]. Thus, it continues to have a central role in the global education and sustainability agenda in the post-2015 era. Numerous attempts are being made at all levels of government to promote and implement this concept at the national and subnational levels [2].

Three basic characteristics of ESD are relevant in the analysis of its governance structure. First, ESD is a concept that has been developed at the international rather than national level and has been recognized and actively promoted by a number of important international organizations [2]. In fact, ESD is genuinely international. Similar to the concept of sustainable development [11], ESD's origins are external to any domestic programme or political discourse. Both sustainable development and ESD were developed and defined by relatively small numbers of actors within the institutional context of the United Nations (UN). To a certain extent, Lafferty's characterization of sustainable development as being an "outside-in obligation" that has evolved "largely outside of the realm of normal domestic politics" [13] (pp. 17–18) can also be applied to ESD.

Second, ESD is a relatively vague concept. Many competing definitions have been proposed (see for example [14]). Some scholars go so far as to challenge even the basic political expectations and ethical assumptions underlying ESD, addressing the term as an empty signifier of the neoliberal logic hidden behind education for so-called ethical behavior (see, e.g., [4,15–19]). The contested nature of the concept manifests itself also in a fragmentation of the terminology with different authors suggesting alternative wordings such as education for sustainability (EfS) or environmental education for sustainability [20–22]. This vagueness of the ESD concept allows UN member states to fill it with their own domestic policy priorities, thereby further increasing the nebulousness of its regulatory content. For example, while in Western countries, ESD often refers to sustainable energy consumption or renewable energies, developing countries more often focus on securing basic living conditions [4]. As a result, and despite being treated as a primary goal of environmental and educational policymaking, ESD is very inconclusive with respect to specific policy prescriptions.

Third, and as a consequence of the first two characteristics, ESD constitutes a relatively "soft" mandate for change. Without clearly defined regulatory prescriptions and being promoted mainly by international organizations without any "hard" sanctioning power, implementation is largely left to the discretion of UN member states. As a consequence, and despite the strong engagement of UNESCO and other intergovernmental organizations, ESD still struggles to find its place within mandatory school curricula in many countries [23]. These three characteristics also apply to the subfield of CCE which constitutes the empirical case presented in Section 4. The "special nature" of ESD and CCE has implications for the global governance structures that have evolved around them. Instead of a hierarchical structure of top-down policymaking, the global ESD and CCE policy domains are characterized by a non-hierarchical network structure. In these networks, a wide range of state and non-state actors, including international organizations, national and sub-national governments, schools, and other educational institutions, as well as professional associations and education-related NGOs cooperate in their efforts to define, concretize, promote, and implement the two concepts [2]. Educational actors are actively searching to build coalitions in order to find ways of operationalizing ESD and CCE by developing common priorities, pedagogical principles, and evaluation procedures. To give two examples, schools have formed alliances with environmental education centers in order

to develop educational programmes. Environmental NGOs are advancing partnerships in early childhood care by offering supplies or solutions for enhancing ESD initiatives in kindergartens [4].

## 3. Using a Network Perspective to Study ESD

Due to the above characterized network structure, applying a SNA perspective in the analysis of the ESD governance structure can offer important insights. During the last decade, SNA has become increasingly prominent in the social sciences [24]. Rather than a single method, SNA constitutes a collection of different quantitative and qualitative approaches which have been developed over many years. There are, however, some general assumptions which unite different network approaches and may be regarded as the basic principles or theoretical fundamentals of the network perspective. For instance, SNA is based on the conception that a network consists of nodes (e.g., individuals or countries) and ties (e.g., interactions or social relations). Whereas most traditional methods of social and political analysis emphasize individuals (or nodes) and their attributes, such as, age, resources, and official role, the primary focus of SNA is on the ties between the nodes and the relational structures in which the nodes are embedded. Hence, it is assumed that the structure of a network influences its performance as a whole, but also the characteristics and capabilities of individual nodes [25]. This stronger focus on the relationships in which actors are embedded constitutes a critical change in how social science attributes influence and power to actors.

Semi-structured interviews and participant observations are traditional methods of data collection used to address questions of influence, however, they might lead to biased results. Being based on statements of the interviewees themselves, or on the direct observation of actors and their strategies, their accuracy depends crucially on the willingness of actors to publicly disclose their preferences. If actors are unwilling to disclose their true preferences and strategies, as often occurs with actors in the education sector [4], empirical results will be flawed. Hence, empirical findings based only on interviewees' statements about the possible influence of specific actors within an issue area or on direct observations of actors' strategic behavior might be under- or overstated. Using SNA techniques in combination with new data sources, such as Twitter data, can overcome these issues.

The most common indicators used to observe influence through SNA are descriptive centrality measures. Degree centrality is the simplest measure, which, in an undirected network, represents the number of ties or social relations of any given node. This measure indicates how well an actor is connected. In directed networks, degree centrality is separated into in-degree centrality (the number of ties directed towards an actor) and out-degree centrality (ties directed from an actor). The former can be interpreted as the relative popularity of a node while the latter indicates a node's potential to act as a multiplier within a network. Other important centrality measures include betweenness and eigenvector centrality. Betweenness centrality is a measure that indicates that a node occupies a broker position. It is measured by how often an actor is situated on the shortest path between two other nodes [26]. In a network of information flows, for example, a high value of betweenness centrality indicates the possibility to partially control the content of the information that is communicated in the network. Eigenvector centrality measures how well an actor is connected to other central actors. However, a high eigenvector centrality score does not necessarily imply many connections. Instead of the pure number of relations, it is the quality of the connections that is decisive [27,28]. Depending on how influence is conceptualized, different indicators can be considered. For example, if an actor is mentioned a lot in Twitter networks around ESD, and thus has a high in-degree centrality value, this actor seems to be very popular in this network. This can be an important decision maker who other actors try to influence by putting them under public pressure through mentioning them in their tweets.

In addition to descriptive measures, inferential SNA techniques have been, and are still being developed to enable hypothesis testing with network data. Network specific methods are needed because the central assumption in regression analysis—that observations are statistically independent and identically distributed (i.i.d.)—are violated with network data. Dependencies of networks need to be included in the models because they might be the main interest of analysis [29]. A family

of statistical models that is used to make inferences from network data are exponential random graph models (ERGMs). In ERGMs, the observed network is treated as an outcome and the aim of the analysis is to identify the data-generating process behind a given network [30]. This means that the covariates (exogenous effects) and the network structures (endogenous effects) are modeled explicitly in an ERGM, and it is tested, whether the observed patterns occur more often than they would in any random network [31]. It is also possible to analyze panel data with inferential SNA techniques, for example with stochastic actor-based models (SAOMs). These models are one of the main options to answer questions about changes in network structures and tie formation over time. The SAOM assumes that tie formations are actor driven, which makes them especially useful to test hypotheses about how actors change their outgoing ties [32]. Analyzing panel data is also possible with an extension of ERGMs, the temporal ERGM [33,34] (for a discussion about the differences between the two approaches, see [35]).

Discourse network analysis (DNA) offers the possibility to analyze network data in a more qualitative way. This approach is a combination of network analysis with categorical content analysis [34]. It overcomes the issue of analyzing either the actor or the frame level of a political discourse and instead integrates both levels into one analysis [36]. The results of the DNA can be visualized through two-mode networks, which include two types of nodes—actors and concepts—within one network. Other transformations can generate, for example, actor congruence networks, concept congruence networks, or conflict networks. DNA allows to trace framing processes within a debate, and to identify the advocacy or discourse coalitions underlying different frames. In the context of ESD, DNA can be used to identify coalitions which are promoting different concepts and framings, and the most important actors within this process.

Hence, a wide range of SNA measures and techniques have been developed and are being deployed. Whereas descriptive quantitative SNA measures have reached maturity, inferential SNA techniques and qualitative SNA approaches present dynamic fields with many ongoing developments.

## 4. Twitter Data as a Source for SNA

Various data sources can be used to conduct SNA. For DNA, newspaper articles are often a useful source for identifying concepts and the actors that promote or oppose them. When using quantitative approaches of SNA, survey data are commonly used. More recently, digital data from OSNs have become popular among social science researchers. Due to the rapid increase of social media users, data from platforms such as Twitter, Facebook, and Instagram are abundant and offer new insights into social interaction and communication. Three billion people are estimated to use OSN by 2021—an increase of 20% from 2.4 billion in 2017 [37]. For network researchers, these data sources are particularly interesting because of the relational nature of the data. OSN data directly provide information about an actor's friends, followers, and interactions, making it easier to build datasets for network analysis.

### 4.1. The Case for Twitter Data

Twitter is an exciting data source for social science researchers for various reasons. The company was founded 13 years ago in 2006 and has since evolved to be one of the biggest OSNs. It provides its users with a platform to communicate with each other through short messages of up to 280 characters called tweets. Data produced through tweets is plentiful. Every day, around 500 million tweets are sent [38] by users around the world. Out of a total of 330 million Twitter users, 262 million are located outside of the United States [39]. Another important aspect for researchers is that interactions take place mostly publicly and therefore are more accessible than data from other OSN [40]. Moreover, the observed interactions occur naturally and in real-time. Thus, it is possible to trace debates over time and to identify the actors framing them. This is distinct from other sources, such as interviews, where interviewees are aware that they are being studied, and thus might adjust their statements or be unable to recall their earlier preferences or actions correctly. For network researchers, another important advantage of Twitter data is that it provides complete networks. This is often not the case

with survey data where non-respondents create missing data, which is problematic for using centrality measures [41,42], and for making inferences [43,44].

The content of tweets also presents an important source of information. Twitter has increasingly become a platform that many political actors use to communicate with each other and with the general public (e.g., [45–47]). In 2018, 187 governments and heads of state were represented on Twitter [48]. During political events, such as UN negotiations or elections, Twitter has become a crucial source for live updates, through so-called live tweeting. This offers a large amount of data on important and otherwise difficult to access persons.

## 4.2. Obtaining Twitter Data

Twitter data can be obtained either through scraping or purchasing. The first possibility, scraping Twitter data yourself, comes with some restrictions. One of them is that historical data can only be obtained for the last seven days for free. Moreover, it is unclear whether the data set will be complete and what kind of algorithm is used to select the obtained tweets [49–51]. Hence, for a complete data set, or for historical data over a longer period of time, buying Twitter data is the better option. This can be done either through Twitter directly or through a third-party vendor. The latter option has become more and more restricted, leaving only a few vendors that primarily focus on selling data to businesses instead of researchers [52].

An important step before purchasing or scraping data is to set up search queries according to which data should be filtered. These search queries can contain hashtags, words and word combinations. On the topic of CCE, hashtags that are frequently used are, e.g., #CCE, #education, and #SDG4. Word combinations are important, for example, when data related to a specific event is needed. Most of the big events have a specific hashtag, e.g., #COP23 for the 23rd global climate conference, the so-called COP (Conference of the Parties). If data related to both this conference and the topic of CCE is to be collected, a combination of the hashtags #COP23 and #education or #COP23 and #CCE might yield useful results.

For some research questions, it can also be useful to gather data from specific accounts only. On a global level, the most important actor on the topic of ESD is UNESCO. This UN organization also maintains various Twitter accounts in different languages. On its main account only, it has over 3 million followers, as of June 2019 and has posted almost 23 thousand tweets [53]. The most important proponent of CCE is the UNFCCC secretariat [54]. For national debates and on the ground implementation of ESD and CCE, local accounts of state and non-state actors are important data sources as well.

## 4.3. From Data to Networks

In this section, we illustrate how to conduct SNA with Twitter data on the topic of CCE. We chose the topic of CCE as a case study in order to exemplify the usefulness of SNA as a tool for analyzing the governance structure related to ESD more generally. Networks based on Twitter data can be constructed in three main ways, depending on the type of interaction [55,56]. The simplest form of interaction is to mention another Twitter user within a tweet with an @-sign (see Figure 1). This tweet is then shared with the followers of the sender and will be visible on its profile page as well. In the example in Figure 1 the sender is @UNESCOEU and the target is the @UNESCO account. Another possibility is to retweet another user's tweet, which means that the message is shared with all followers, either in its original form or with an additional comment of the person retweeting it. The third option is to reply to a tweet. In this case, the user to whom the reply is directed will be mentioned below the original post with an @-sign, and the reply is also shared with the sender's followers. These three different forms of interaction represent the ties in a network. Twitter users are visualized as nodes.

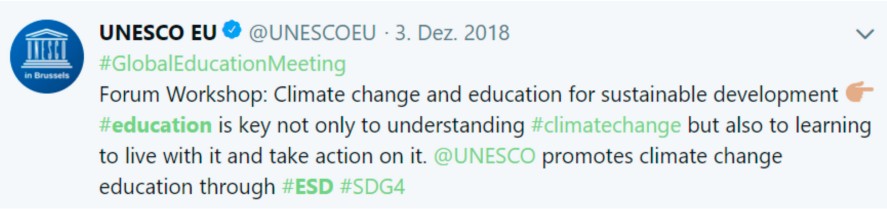

**Figure 1.** An exemplary tweet from UNESCO EU (the United Nations Educational, Scientific and Cultural Organization's Representation to the European Institutions). Source: [57]

Depending on the research question, the direction of these interactions can be modelled differently. If the research goal is to analyze information flows between actors involved in the ESD policy domain, i.e., to disclose who spreads information on Twitter on the topic of CCE, the direction in a retweet would be from the user of the original tweet to the one who retweeted. However, if the goal is to analyze who confers visibility to whom on Twitter, the direction would be modelled the other way around—from the user who retweeted the one who posted the tweet originally. In our illustrative SNA, we will conceptualize the interaction direction according to the second option presented. Our aim is to identify the actors who are most influential within CCE-specific debates, applying a concept of influence in which actors who receive a lot of attention through any kind of interaction (mentions, retweets, or replies) exert influence over the debate.

As the basis for our analysis, we chose to use data collected during the annual global climate summits. Taking place at the end of every year, these negotiations last for around two weeks each. Education is a crucial part of the COPs. From the beginning of the UNFCCC regime in 1992, when the convention was first adopted [58], to the latest commitment of the parties with the Paris Agreement of 2015 [59], education and its role with respect to addressing climate change was included. During recent years, the topic of education has experienced increasing attention. Since COP21 there has been an education day every year, and actions around education are referred to as "Action for Climate Empowerment" (ACE), instead of "Actions under Article 6 of the Convention" [54].

In this analysis, we examine Twitter data collected during these negotiations over the three-year period from 2015 to 2017 (or from COP21 to COP23). We obtained our data from a former third-party vendor called Texifter. As mentioned earlier, many third-party vendors are not selling Twitter data anymore which is also the case for Texifter [60]. In our first search query that we created to communicate our request to Texifter, we used the number of COPs as the main indicator, as well as words and word combinations that our research focuses on. Education was one of these topics. In a second step, we needed to reduce the extensive amount of data we obtained to interactions in order to be able to create networks. Hence, tweets that did not contain interactions of any kind were excluded from our data set. We kept all tweets that included interactions of either type: Mention, retweet, or reply. The remaining data were filtered according to the topic of education, our main research interest for this study. Choosing the global climate summits as a time frame for our analysis, we assume that all education-related tweets are CCE-specific. To receive all education-related interactions during the three COPs, we used a search query including the following words and hashtags:

> education OR educators OR EduDay OR education day OR ClimateChangeEducation OR climateeducation OR climate education OR ESD OR education for sustainable development OR #SDG4 OR #ACEnow OR #GAPesd OR #ACE

The created networks vary significantly in size. COP21 was the most important conference of the three, because the Paris Agreement was adopted in that year and therefore, in general, more tweets were sent during that summit. While during COP21, we received more than four million tweets from Texifter, for COP22 and COP23 we obtained around one million tweets each. This difference in the overall activity during these summits is also reflected in the sizes of the CCE-specific networks. COP21 has by far the biggest network with 9,475 ties and 5,313 nodes. The COP22 network is almost half as

big with 5,728 ties and 3,268 nodes, and the one for COP23 is the smallest CEE-specific network with 3,895 ties and 2,218 nodes.

After creating the networks with the software Gephi, we applied the measure of PageRank to identify the most influential actors. PageRank is a measure similar to eigenvector centrality (see Section 3). Whereas eigenvector centrality usually applies degree centrality to measure influence, PageRank uses in-degree centrality and also accounts for the weights of the ties [61]. We found this measure to be the most appropriate for our conceptualization of influence. Table 1 provides an overview of the 15 actors with the highest PageRank values in each network and Figure 2 depicts the networks.

**Table 1.** Actors with the highest PageRank values.

|  | COP21 | COP22 | COP23 |
|---|---|---|---|
| 1. | UNFCCC | UN | UNFCCC |
| 2. | UNESCO | UNFCCC | PEspinosaC |
| 3. | UNICEF | UNESCO | UN |
| 4. | najatvb | BofA_News | Connect4Climate |
| 5. | COP21en | GBLFoundation | COP23 |
| 6. | haliscolb | GEMReport | BMZ_Bund |
| 7. | IrinaBokova | HuffPostGreen | UNESCO |
| 8. | RoyalSegolene | COP22 | CLIMATEwBORDERS |
| 9. | UNICEFtalk | ManosAntoninis | uncclearn |
| 10. | COP21 | IrinaBokova | larutadelclima |
| 11. | UNITAR | Education2030UN | SeruiratuCOP23 |
| 12. | UNEP | PEspinosaC | COP22 |
| 13. | UNwomen | UNEP | RisingSign |
| 14. | earthguardianz | Abibimman | unescoNOW |
| 15. | unicefniger | ClimateCoLab | OECD |

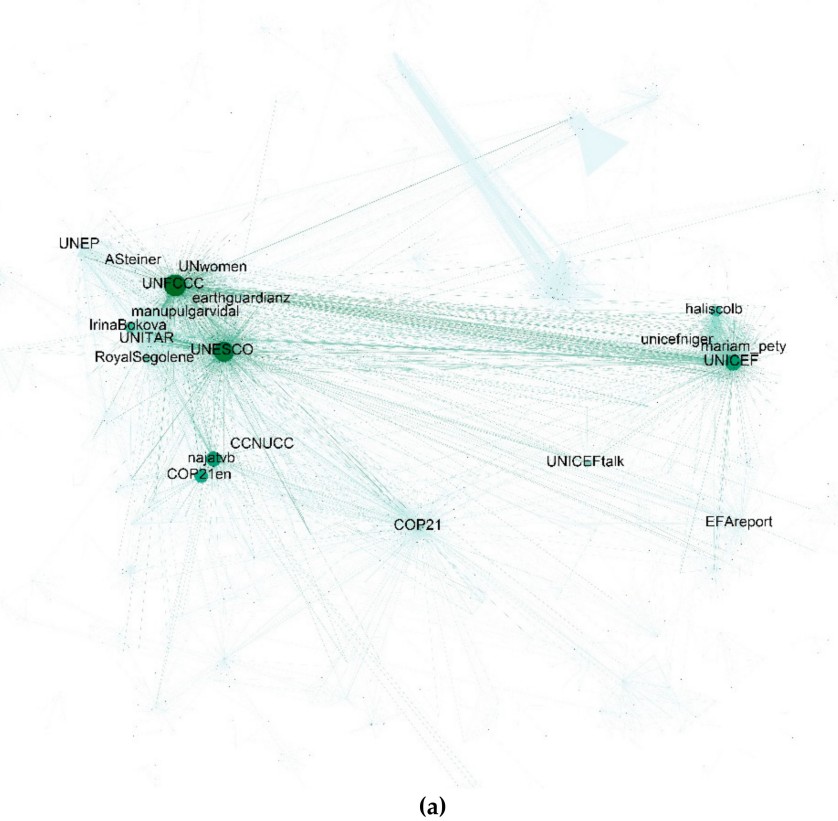

**(a)**

**Figure 2.** *Cont.*

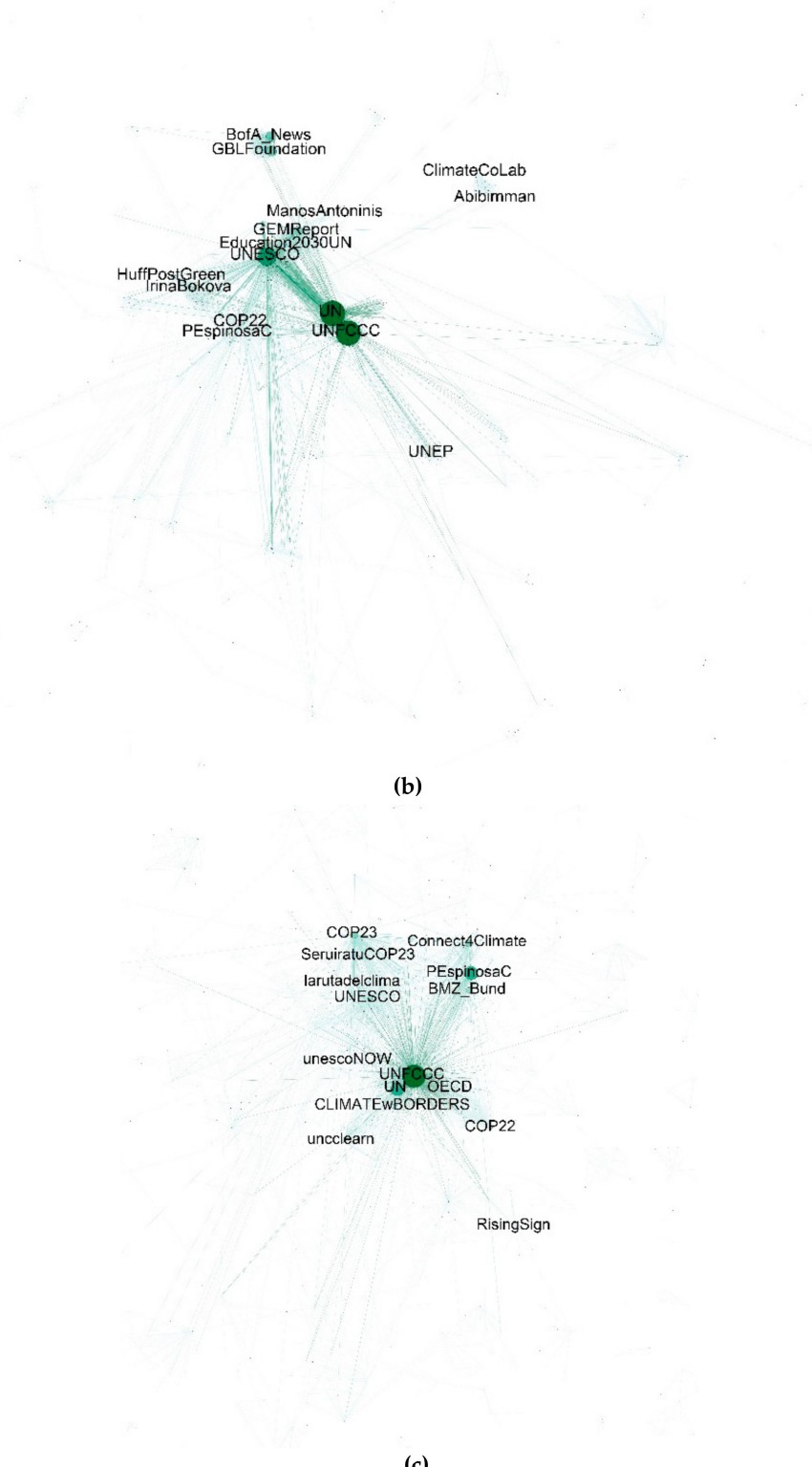

**Figure 2.** CCE-specific network during COP21-23 on Twitter. (**a**) COP21 network (**b**) COP22 network (**c**) COP23 network. The networks were created with Gephi using the OpenOrd layout. The color and the node size represent the PageRank values. To keep the networks legible, single nodes on the margins were excluded.

Across all three years, accounts of IOs, such as UNESCO, UNICEF (United Nations International Children's Emergency Fund) and of the international treaty secretariat of the UNFCCC are the ones

with the highest PageRank values. Individuals working for these organizations, such as Patricia Espinosa (@PEspinosaC), who has been the Executive Secretary of the UNFCCC since 2016, or Irina Bokowa (@IrinaBokova), who was Director-General of UNESCO until 2017, are also among the most influential actors.

Apart from the fact that these actors received the highest PageRank scores, international organizations, moreover, are represented the most in each year. During COP21, ten Twitter accounts belonged to IOs or their representatives: @UNFCCC, @UNESCO, @UNICEF, @IrinaBokova, @UNICEFtalk, @UNITAR (United Nations Institute for Training and Research), @UNEP (United Nations Environment Programme), @UNwomen (UN agency working for gender equality & women's empowerment), and @unicefniger (UNICEF in Niger). The account of @haliscolb belongs to one of UNICEF's Youth Ambassadors during COP21 and was also counted as part of the IO group.

At COP22, the overall structure looks very similar. Nine out of the fifteen most influential Twitter accounts belong to IOs: @UN, @UNFCCC, @UNESCO, @GEMReport (UNESCO Global Education Monitoring Report), @ManosAntoninis (Director of the Global Education Monitoring Report), @IrinaBokova, @Education2030UN (Twitter account for Sustainable Development Goal 4 "Quality Education"), @PEspinosaC, and @UNEP. During COP23, seven international organizations, their representatives or their initiatives, are among the most influential accounts: @UNFCCC, @PEspinosaC, @UN, @UNESCO, @uncclearn (UN Climate Change Learning Partnership), @unescoNOW, and @OECD (Organization for Economic Cooperation and Development). Of those IOs represented, UNESCO and UNFCCC are the most influential across all years. Interestingly, some IOs only score high PageRank values in one year, such as @UNICEF at COP21 or @OECD at COP23. This might lead to the suggestion that they played a more important role at that time, for example through organizing events or launching publications on the topic of CCE.

Other highly influential actors are the hosts of the annual summits, such as the French (@COP21), Moroccan (@COP22), and Fijian presidencies (@COP23). These COP accounts are officially managed by the respective country that holds the presidency of that year but receive support from the UNFCCC secretariat. We expect them to be influential due to the nature of their role, and thus their influence is unlikely to be limited to CEE-specific debates. Moreover, national ministries of the host countries also hold central positions within CCE debates during the COPs on Twitter. The French Minister for Education, Najat Vallaud-Belkacem (@najatvb) and the Minister for Environment, Ségolène Royal (@RoyalSegolene) were both influential actors with relatively high PageRank values during COP21. Interestingly, during COP23—when the Republic of Fiji held the presidency, but for logistical reasons, the summit was hosted in Germany—instead of the education or the environment ministry, the German Ministry of Economic Cooperation and Development (@BMZ_Bund) was the most influential ministry within CEE-specific debates.

Various civil society organizations (CSO) also occupy central positions within the networks. During COP21, Earth Guardians (@earthguardianz) was the only CSO ranked as one of the top 15 influential actors. During COP22, three CSO made the top 15 list: The Global Bright Light Foundation (@GBLFoundation), the Abibimman Foundation (@Abibimman), and the ClimateCoLab (@ClimateCoLab). In 2017, during COP23, Climate Without Borders (@CLIMATEwBORDERS) and La Ruta del Clima (@larutadelclima) were the most influential CSOs. Most of these organizations focus on topics directly associated with CCE, such as youth, education, and climate change (@earthguardianz, @larutadelclim, @Abibimman, @CLIMATEwBORDERS, @ClimateCoLab). The Global Bright Light Foundation is the only CSO that is working on the topic of energy.

Hence, our SNA results based on PageRank values show that IOs are in fact the ones who exert the most influence over CEE debates during global climate summits, and thus, form an essential part of the governance structure. Using the measure of PageRank for a descriptive SNA is one possibility to identify influential actors within Twitter networks. Depending on the research interest, however, other measures could be more appropriate.

### 4.4. Limitations of Twitter Data

Twitter data as a source for SNA have many advantages, such as accessibility and the observation of real-time and natural interactions. However, as with other data sources, there are some possible drawbacks. The most obvious caveat is that not all actors relevant for a research question have Twitter accounts. Although Twitter has become an important tool for many people and organizations to interact with each other or state opinions and preferences, not everyone uses the OSN. In the case of CCE and ESD, this might be especially challenging, when the research interest is based on the implementation of the two concepts at a local level. Actors that are primarily responsible for the implementation on the ground, such as local governments and schools, might not be represented, and thus crucial actors would be left out of the analysis.

Another possible bias of Twitter data comes with its public nature. For many researchers, this is the appeal of Twitter data. However, the connections maintained on Twitter do not necessarily represent social relations that occur beyond the platform. Similarly, information shared on Twitter might differ significantly from the information shared privately. Thus, Twitter data is an additional source of information that does not necessarily mirror interactions that occur in a private environment.

Moreover, the information that Twitter provides is fixed. It is limited by the data offered through the tweet content of 280 characters and Twitter's metadata. Although metadata has been found to be quite powerful (e.g., [62]), it is predetermined by the platform provider, not the researcher. This is different for survey and interview data, where researchers can pose exactly the questions they need to gather information for their research goal.

For longitudinal analyses, Twitter's fast-paced nature poses an additional challenge. Accounts on Twitter get deleted, and users become inactive or change their names. A central issue for longitudinal SNA are accounts that only exist in certain years (see Section 4.1). Reducing the data set to only those actors that are active during all years of the analysis is one possibility to deal with this issue. Another option can be to impute data that is missing. The issue of varying usernames can be solved by using IDs of the accounts instead of usernames.

Twitter data is a promising source for social science researchers when applied appropriately. Instead of providing an alternative to traditional data sources, Twitter data should be considered as a useful addition. In combination with other data sources, it provides a more comprehensive analysis of many research questions.

## 5. Discussion and Conclusions

Networks of state and non-state actors have been identified as the dominant structures in the governance of ESD and CCE. This structure can be attributed to the three basic characteristics of the ESD and CCE concepts laid out in Section 2: The international origin of both concepts, their conceptual ambiguity, and the limited sanctioning power of the international organizations that promote them. Against this background, we argued that SNA offers adequate concepts and techniques to examine the governance structures around ESD and CCE in more detail. In the study of governance structures, the relational focus of SNA provides us with important possibilities to attribute influence or power in an indirect way. This is different from traditional methods that base their results on statements of actors themselves or on the direct observation of these actors' behavior. Moreover, we argued that Twitter data present an interesting basis upon which to examine governance structures, which has not yet been exhausted.

To illustrate our argument, in this article, we conducted an SNA based on Twitter data on the topic of ESD. Analyzing Twitter data during climate change summits over three years (2015–2017), we showed how SNA measures can be used to identify influential actors in debates around CCE, which we conceptualize as a subfield of ESD. To measure influence within CCE-specific networks on Twitter, we applied the PageRank algorithm. Similar to the eigenvector centrality measure, it assesses how well an actor is connected to other well-connected actors in directed networks. Our results show that international organizations, such as UNESCO and UNICEF, as well as the UNFCCC

secretariat, are the actors with the highest PageRank values across all three years of analysis. Moreover, international organizations are also the ones that appear most often among the top 15 actors (see Table 1). This means that international organizations are the most central actors within CEE specific debates and therefore have the potential to exert significant influence within these governance networks.

These findings are in line with our initial observation of the international origin of the concepts of ESD and CCE. However, they go further by suggesting that IOs were not only influential in developing these concepts, but are also among the most central actors in the subsequent process of interpretation, further development, and implementation of ESD and CCE. Our finding that IOs were the most frequently mentioned group of actors within the issue-specific debates that occurred around COPs 21 to 23 suggests that they continue to shape the concept of CCE by presenting reports and hosting events on the topic which reach a wide range of actors during these summits.

Other groups of actors that have been identified as influential in our analysis are nation states, particularly the accounts of the respective COP presidencies and national ministries concerned with the topic of CEE, as well as civil society organizations. The influence of different nation states and some of their ministers could be attributed to their roles as host countries during these global climate summits. Their influence is, however, unlikely to be limited only to the topic of CCE. Among the most influential CSOs, organizations that focus on youth empowerment, climate change, and education are predominant.

Our analysis demonstrates that SNA based on Twitter data offers untapped possibilities to analyze governance structures that form around policy issues such as ESD and CCE. Information provided on Twitter is not detached from political reality. Rather, they add an additional layer to it. OSNs such as Twitter offer possibilities to state actors and IOs to communicate with other political actors, but also to interact with citizens directly and publicly. Twitter constitutes a particularly political platform in this regard. Other big OSNs such as Facebook, Instagram, and Snapchat are not as politicized in the sense that the main function of these platforms is to exchange personal information, whereas Twitter provides the greatest opportunity to participate in public debates. This makes Twitter especially interesting for political science researchers.

Although Twitter data offer interesting possibilities, its usefulness might be restricted to certain actors at certain times. Defining policy issues and framing them is the part of the governance process where Twitter can be extremely helpful for political actors. However, when it comes to the implementation of policies and regulations, Twitter might not be the most relevant platform for actors to exchange information on these processes. As we showed in this illustrative case, IOs, the organizations defining the concepts, constituted the most central group of actors in the networks. Local actors, who are extremely important for the implementation, however, did not appear as central. Hence, researchers interested in the implementation processes of the concepts should rather conduct surveys or interviews with local actors to examine local governance structures [2]. These surveys and interviews could also be used in combination with SNA techniques.

To conclude, in this study, we used one specific centrality measure to analyze the network governance structure of ESD and more specifically CCE. The measure of PageRank is only one possibility to conduct SNA. Other centrality measures might provide additional insights on the governance structure. For example, applying the measure of betweenness centrality would allow to identify actors that are potential brokers within the CEE-specific networks. SNA also offers approaches beyond descriptive centrality measures. Future research might, for example, use DNA to examine the use of different framings by groups of actors, thereby identifying coalitions that form around specific framings of ESD and CCE. Moreover, inferential SNA techniques could be used to identify the attributes (e.g., actor type and country) and network structures that make ties between actors more likely. The appropriate measure and SNA approach is thus dependent on the primary research interest. Our results can be considered as a basis for future SNA which create a more thorough understanding of the governance structures around ESD and CCE.

**Author Contributions:** This study is part of a research project that was acquired and is supervised by N.K. and H.J. Both contributed in the conceptualization process, the preparation of the original draft as well as in the review and editing process. A.G. managed the data, conducted the analysis and visualized the networks. She was also part of the conceptualization process, the original drafting as well as the review and editing process.

**Funding:** Our research project is part of the research unit "International Public Administration," funded by the German Research Foundation through the grants KO 4997/1-1, KO 4997/4-1.

**Acknowledgments:** We would like to thank Lea Susanne Helm for her research assistance on this study and Freie Universität Berlin for funding the publication of this article.

**Conflicts of Interest:** The authors declare no conflict of interest. The funders had no role in the design of the study; in the collection, analyses, or interpretation of data; in the writing of the manuscript, or in the decision to publish the results.

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
