# Peer review of "Education for Sustainable Development and Climate Change Education: The Potential of Social Network Analysis Based on Twitter Data"

_sustainability, doi:10.3390/su11195499_

Round 1

Reviewer 1 Report

The manuscript entitled "Education for Sustainable Development and Climate Change Education: The Potential of Social Network Analysis Based on Twitter Data" is original, well structured and clearly presented research work. Very easy to follow and provide sufficient background information. SNA confirmed previous findings:  SD and ESD being very external, disconnected from domestic, national policy and practice.

Overall, very good work. I recommend acceptance of the manuscript after minor revision.

My suggestions for improvements of the manuscript:

Page 3, line 103: Write one or two sentences about GAP and other initiatives after UN ESD Decade 2005-2014. There is already five years since 2014.

Page 3, line 127: I would recommend that you read and add the articles by Selby. They provide very good insight into disconnection between global and local view of SD:

Selby, D., & Kagawa, F. (2018). Archipelagos of learning: environmental education on islands. Environmental Conservation, 45(2), 137-146.

Selby, D., & Kagawa, F. (2018). Teetering on the Brink: Subversive and Restorative Learning in Times of Climate Turmoil and Disaster. Journal of Transformative Education, 16(4), 302-322.

Author(s) described very-well advantages and disadvantages of using Twitter data for SNA. It would be beneficial if author(s) would make some recommendations in the conclusion on how feasible is to use different OSNs in such studies, since not everyone is using them, different target groups could be reached with OSNs, would it be beneficial to combine different research methods to sufficiently analyze the network… ?

Author Response

The manuscript entitled "Education for Sustainable Development and Climate Change Education: The Potential of Social Network Analysis Based on Twitter Data" is original, well structured and clearly presented research work. Very easy to follow and provide sufficient background information. SNA confirmed previous findings: SD and ESD being very external, disconnected from domestic, national policy and practice.

Overall, very good work. I recommend acceptance of the manuscript after minor revision.

We thank the reviewer for this overall positive assessment of our article.

My suggestions for improvements of the manuscript:

Page 3, line 103: Write one or two sentences about GAP and other initiatives after UN ESD Decade 2005-2014. There is already five years since 2014.

Thank you for this comment. We incorporated the current developments with the GAP in lines 108-110.

Page 3, line 127: I would recommend that you read and add the articles by Selby. They provide very good insight into disconnection between global and local view of SD:

Selby, D., & Kagawa, F. (2018). Archipelagos of learning: environmental education on islands. Environmental Conservation, 45(2), 137-146.

Selby, D., & Kagawa, F. (2018). Teetering on the Brink: Subversive and Restorative Learning in Times of Climate Turmoil and Disaster. Journal of Transformative Education, 16(4), 302-322.

We thank the reviewer for providing these literature suggestions. We incorporated the two references in line 128.

Author(s) described very-well advantages and disadvantages of using Twitter data for SNA. It would be beneficial if author(s) would make some recommendations in the conclusion on how feasible is to use different OSNs in such studies, since not everyone is using them, different target groups could be reached with OSNs, would it be beneficial to combine different research methods to sufficiently analyze the network…?

Thank you for these recommendations. We rewrote the conclusion and included another paragraph (line 497-516) about the special character of Twitter as a social network in contrast to other OSNs. We also mentioned possibilities of how to combine data collected through OSNs with other data sources.

Reviewer 2 Report

The authors have written an interesting study, but there are important questions that motivate a new review of the publication:

Line 90: The next section… Which one? Given that the discussion of results in a scientific article shows the real contribution to knowledge, I recommend that the results be interpreted and that the implications of these results be reflected upon. Authors should write the paper discussion using research information that they have collected beforehand. Usually, should showcase each problem individually and impartially, discussing one side and then the other side of each issue regarding the topic. Try to move on through your key arguments in specific order, starting with your weakest argument and gradually progressing to the strongest. This particular structure allows readers to follow the thoughts without getting distracted. When writing the discussion section, authors should carefully consider all possible explanations for the study results, rather than just those that fit your hypothesis or prior assumptions and biases. So, authors should: Discuss your results in order of most to least important; Compare your results with those from other studies; Discuss what your results may mean for researchers in the same field as you, researchers in other fields, and the general public. How could your findings be applied?; State how your results extend the findings of previous studies.

Author Response

Responses to Reviewer #2

The authors have written an interesting study, but there are important questions that motivate a new review of the publication:

We thank the reviewer for this overall positive assessment of our article.

Line 90: The next section… Which one?

Thank you for the comment. We clarified the wording in line 90 and changed it into “section two”.

Given that the discussion of results in a scientific article shows the real contribution to knowledge, I recommend that the results be interpreted and that the implications of these results be reflected upon. Authors should write the paper discussion using research information that they have collected beforehand. Usually, should showcase each problem individually and impartially, discussing one side and then the other side of each issue regarding the topic. Try to move on through your key arguments in specific order, starting with your weakest argument and gradually progressing to the strongest. This particular structure allows readers to follow the thoughts without getting distracted. When writing the discussion section, authors should carefully consider all possible explanations for the study results, rather than just those that fit your hypothesis or prior assumptions and biases. So, authors should: Discuss your results in order of most to least important; Compare your results with those from other studies; Discuss what your results may mean for researchers in the same field as you, researchers in other fields, and the general public. How could your findings be applied?; State how your results extend the findings of previous studies.

Thank you very much for this recommendation. We rewrote the conclusion, stated our arguments more clearly and explained how our findings underpin our arguments. Moreover, we interpreted our findings and the potential meaning within this research field. The limitations of our data and methodological approach were also revised.

Round 2

Reviewer 2 Report

Accept in present form.